# Flexible degrees of connectivity under synaptic weight constraints

**Gabriel Koch Ocker**
Allen Institute for Brain Science
Seattle, WA 98109
gabeo@alleninstitute.org

**Michael A. Buice**
Allen Institute for Brain Science
Seattle, WA 98109
michaelbu@alleninstitute.org

## Abstract

Biological neural networks face homeostatic and resource constraints that restrict the allowed configurations of connection weights. If a constraint is tight it defines a very small solution space, and the sizes of these constraint spaces determine their potential overlap with the solutions for computational tasks. We study the geometry of the solution spaces for constraints on neurons' total synaptic weight and on individual synaptic weights, characterizing the connection degrees (numbers of partners) that maximize the size of these solution spaces. We then hypothesize that the size of constraints' solution spaces could serve as a cost function governing neural circuit development. We develop analytical approximations and bounds for the model evidence of the maximum entropy degree distributions under these cost functions. We test these on a published electron microscopic connectome of an associative learning center in the fly brain, finding evidence for a developmental progression in circuit structure.

## 1 Introduction

Computation in neural networks is constrained by their architecture [14]. The capacity of a network (the number of computations it can successfully learn) depends on a number of factors. For simple associative memory models, the capacity depends on the structure of the inputs [15], the learning rule [26], and constraints on the connectivity [6]. In biological neural networks, the cost function, learning rule, and structure of input activity are often unknown. Increasingly, however, high-throughput connectomics studies are revealing the architecture of neural circuits (e.g., [18, 23, 1, 13, 27, 32]). This allows us to examine biological circuit structures for signatures of developmentally inspired cost functions governing network architectures. Biological circuit structure is shaped by developmental programs and slow structural plasticity, which construct a scaffold for and stabilize learning and memory on faster timescales [22]. Motivated by this, we hypothesize that developmental programs that structure circuits might aim for flexibility: to optimize the number of available weight configurations under given constraints.

The total strength of synaptic connections between two neurons is limited by the amount of receptor and neurotransmitter available and the size of the synapse [19]. Pyramidal neurons of mammalian cortex and hippocampus undergo synaptic scaling, regulating their total synaptic input strengths to stabilize postsynaptic activity levels [29]. We consider simple models of resource limitations and homeostatic constraints on total and individual synaptic weights. We examine how the size of the solution space for these constraints depends on the number of connections (the degree) and compute the optimally flexible degrees under different constraints on total and individual connection strengths. We then develop the maximum entropy degree distributions under these constraints. We derive the Laplace approximation for the evidence of these degree distribution models. Finally, we apply these models to a recently characterized connectome of a learning and memory center of the larval Drosophila melanogaster [13], asking which constraints best explain the degree distributions of neurons at different developmental stages. We find that overall, a homeostatically fixed net weight best predicts the degree distributions of Kenyon cell inputs and outputs. The most mature Kenyon cells, however, are better explained by a simple binomial random wiring

model, suggesting a developmental progression in the cost functions governing mushroom body wiring. Most of the results of this abstract are presented in more detail in a preprint [24].

## 2 Geometry of constraint spaces

We consider a simple model of synaptic interactions where a neuron has degree $K$ and the total strength of projection $i$ is $J_i$. $K$ is the synaptic degree. We assume that the existence of a connection is determined separately from its strength, and model individual synaptic weights as continuous variables, so the weight configuration is a point in $K$-dimensional Euclidean space. Given $K$, a constraint on synaptic weights defines a solution space. We will call the size of that constraint's solution space the flexibility of the constraint. The size of that solution space measures the flexibility of the constraint. are dimensionless quantities and measure the size of a $D$-dimensional solution space by its $D$-dimensional Hausdorff measure (normalized so that the $K$-dimensional Haussdorff measure coincides with the standard Lebesgue measure). For any $K$, the size of the constraint space approximates the number of $K$-dimensional synaptic weight configurations allowed. If the synaptic weights are viewed as a channel dependent on input patterns $X$, the marginal entropy (log size) of the synaptic weights also gives an upper bound on the mutual information between the inputs and outputs (though it may not be a tight bound). We thus consider the constraint's flexibility a simple bound on the computational capacity of a neuron.

**Flexibility under bounded net synaptic weight**    We begin by considering an upper bound on the net synaptic weight, so that

$$\sum_{i=1}^{K} J_i \leq \bar{J} K^p \tag{1}$$

This bound could be interpreted multiple ways, for example as a presynaptic limit due to the number of vesicles currently available before more are manufactured or a postsynaptic limit due to the amount of dendritic tree available for synaptic inputs. Scaling the summed synaptic weight as $K^p$ corresponds to scaling the individual synaptic weights as $K^{p-1}$. If every synaptic weight has an order $1/K$ strength, the sum of the synaptic weights would be order 1 and $p = 0$. If every synaptic weight has an order 1 strength, the summed weight is order $K$ and $p = 1$. If synaptic weights have balanced $(1/\sqrt{K})$ scaling [30], then the summed weight would have $p = 1/2$. We require $0 \leq p \leq 1$ and $\bar{J} > 0$.

With degree $K$, the solution space for Eq. 1 is the volume under a $K - 1$ simplex (Fig. 1a). Thus, for the bounded weight constraint the number of weight configurations is proportional to the volume of the $K - 1$ dimensional simplex, $V(K, \bar{J}) = \left(\bar{J} K^p\right)^K / K!$ (Fig. 1b). We can also view this as a count of available configurations if we divide the maximum synaptic weight $\bar{J}$ into $N$ increments of measurable synaptic weight changes $\Delta J$, and measure synaptic weights relative to $\Delta J$ [24]. In the continuum limit $N \to \infty$, $\Delta J \to 0$ with $\bar{J}$ fixed, the volume under the simplex approximates the number of synaptic weight configurations. We call the synaptic degree that maximizes the volume under the simplex the optimal degree, $K^*$. We computed this optimal degree [24]. It is approximately linearly related to the total synaptic weight:

$$(K^*)^p \bar{J} = \left(K^* + \frac{1}{2}\right) \exp\left(-p\right) + \mathcal{O}\left(1/K^*\right) \tag{2}$$

with a slope that depends on $p$ (Fig. 1c). We can see from Eq. 2 that if $p = 1$, we obtain the condition $\bar{J} = 1/e$ (to leading order). So if $p = 1$ and $\bar{J} = 1/e$, the volume is approximately independent of $K$. If $p = 1$, the volume decreases monotonically for $\bar{J} < 1/e$ and increases monotonically for $\bar{J} > 1/e$.

**Flexibility under fixed net synaptic weights**    Motivated by the observation that different types of neuron regulate their total synaptic weights [29], we also consider a simple model of homeostatic synaptic scaling: $\sum_{j=1}^{K} J_j = K^p \bar{J}$. The fixed net weight constraint defines the same simplices as the bounded net weight, but requires synaptic weights to live on their surfaces instead of the volumes under them (Fig. 1d). The size of this space of allowed weights is given by the surface area of the $K - 1$ simplex, $A(K, \bar{J}, p) = \left(K^p \bar{J}\right)^{K-1} \sqrt{K} / (K-1)!$. The surface area of the simplex increases with the net excitatory weight, but for $\bar{J} \geq 1$ it has a maximum at positive $K$ (Fig. 1e). The optimal degrees obey [24]:

$$(K^*)^p \bar{J} = (K^* + p - 1) \exp\left(-p\right) + \mathcal{O}\left(1/K^*\right) \tag{3}$$

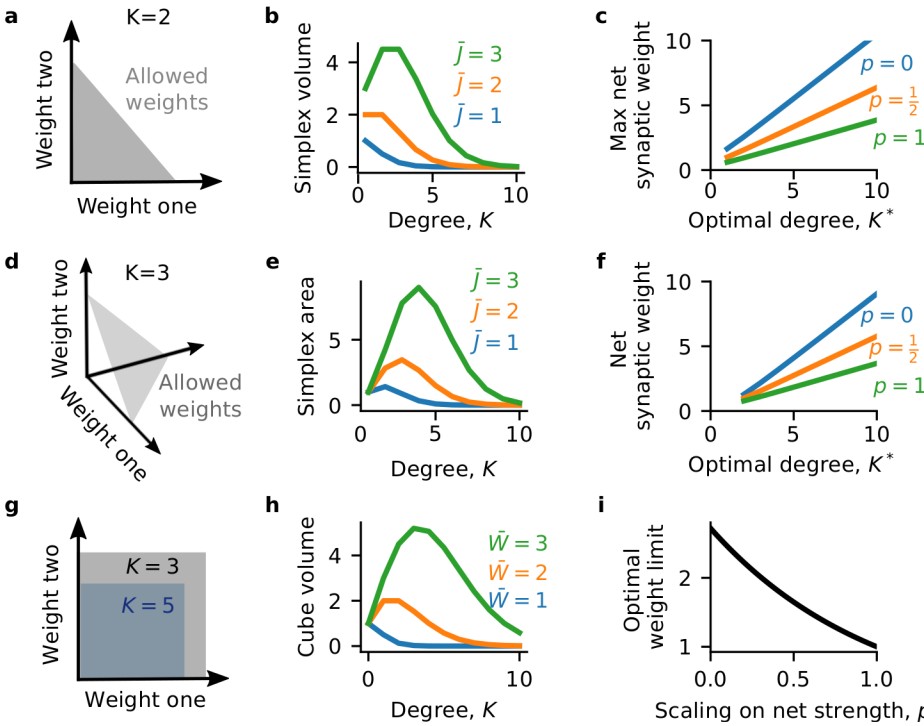

Figure 1: **Constraints on synaptic connectivity.** (**a**) The set of allowed synaptic weights under a bounded net weight for two inputs. (**b**) Volume of the $K-1$ simplex as a function of $K$ (here with $p=0$). (**c**) Relationship between the degree that optimizes the volume under the $K-1$ simplex and the maximal net synaptic weight. (**d-f**) Same as (**a-c**) for the fixed net weight constraint. (**g, h**) Same as (**a, b**) for bounded individual synaptic weights. (**i**) Relation between the scaling of the net synaptic weight, $p$, and the limit on individual synaptic weights at the maximum of the hypercube's volume.

revealing an approximately linear relationship, similar to the constraint on the maximum possible synaptic weight (Eq. 2). As for the bounded net weight, we can see from Eq. 3 that if $p=1$, we obtain the condition $\bar{J}=1/e$ (to leading order). So if $p=1$ and $\bar{J}=1/e$, the surface area is approximately independent of $K$. If $p=1$, the area decreases monotonically for $\bar{J}<1/e$ and increases monotonically for $\bar{J}>1/e$.

**Flexibility under individual connection strength bounds** We consider a simple model for resource limitations at individual connections: $J_j \leq \bar{W}K^{p-1}$. The scaling with $K^{p-1}$ here ensures that the sum of $K$ synaptic weights scales as $K^p$, as for the previous constraints. The volume of the hypercube, $C=\left(\bar{W}K^{p-1}\right)^K$ measures the size of the solution space. If $p=1$ here (individual synaptic weights do not scale with $K$) then the volume of the cube only decreases with $K$ for $\bar{W}<1$ and increases with $K$ for $\bar{W}>1$. If $p<1$, however, the volume exhibits a maximum at positive $K$ (Fig. 1h). At those maxima,

$$\bar{W}(K^*)^{p-1} = \exp(1-p) \tag{4}$$

In contrast to the constraints on the total connection strength, the upper limit for each connection strength is independent of $K$. In all these cases, the value of the constraint at the optimal degree decreases with $p$.

## 3 Maximum entropy degree distributions under connectivity constraints

We next asked what connectivity these different constraints predict. For given $\bar{J}$, $p$, and $K$, the maximum entropy distribution on the synaptic weight configurations $J$ under that constraint is the uniform distribution over its solution space, $\mathcal{S}_K$. For the bounded net weight (Eq. **??**), for example, $\mathcal{S}_K$ is the volume under the $(K-1)$ regular simplex with vertices at $\bar{J}K^p$ (Fig. **??**a). We assume that a developmental process chooses $K$ without respect to the weight configurations so that for $K$ from 1 to some finite maximum $K^{\mathrm{max}}$, the maximum entropy distribution for synaptic weight configurations $J$ is uniform over the union

of $\mathcal{S}_1, \ldots, \mathcal{S}_K$. In this case, the degree distributions are proportional to the size of the solution space:

$$p\left(K \mid \bar{J}, p\right) = |\mathcal{S}_K|/Z_\mathcal{S} \tag{5}$$

where $|\mathcal{S}_K|$ is the size of the solution set $\mathcal{S}_K$ (for the bounded net weight, the volume under the simplex and for the fixed net weight, the surface area of the simplex). For large $K^{\max}$, we compute the normalization constant as $Z_\mathcal{S}\left(\bar{J}, p\right) = \sum_{K=1}^{K^{\max}} |\mathcal{S}_K|$. These provide predictions for neural degree distributions. To test them we turned to an electron microscopic reconstruction of connectivity of Kenyon cells in the larval Drosophila melanogaster's mushroom bodies, a center of learning and memory in the fly brain [13]. These data include the number of synapses for each connection. To map these anatomical measurements onto our theory, we assume that the synapse counts are proportional to the physiological synaptic weights (for constraints on the net synaptic weight, $\bar{J}K^p = \alpha \bar{S}$). We computed the Laplace approximation for the model evidence (marginal likelihood) under each of the constraints discussed above, marginalizing out the scale factor $\alpha$ relating the anatomical measurements to the modeled synaptic weights [24]. We also computed the model evidence for a binomial random wiring model, using anatomical estimates for the number of potential partners of Kenyon cells [24].

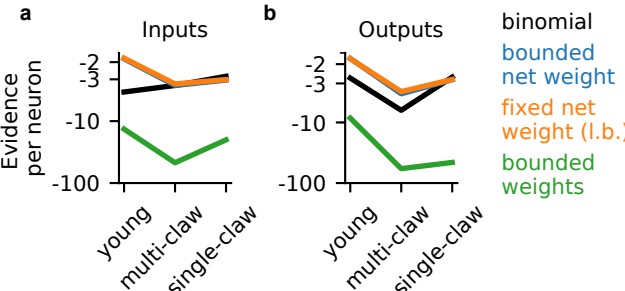

Figure 2: **Mushroom body connectivity.** (**a**) Model evidence for Kenyon cells' in-degree distributions. Young, multi-claw and single-claw Kenyon cells are in order of increasing developmental maturity. The model evidences for the fixed net weight model is a lower bound arising from a bound on its normalization constant. (**b**) Same, for Kenyon cell out-degrees.

KCs can be morphologically classified by the structure of their dendrites. Immature KCs have smooth dendrites, while more mature KCs' dendrites exhibit claws around input axons. Single-claw KCs are more mature than multi-claw KCs [13]. For models with a fixed net weight, the normalization constant was not tractable; because of this we computed bounds for the evidence (Fig. 2b, c orange shows the lower bounds; see [24] for more details).

The models with bounded individual synaptic weights provided the poorest explanations for KC connectivity degrees (Fig. 2b, c green). For single-claw KCs, the binomial wiring model had the highest evidence (Fig. 2b, c black; log likelihood ratio at least 1.57 for binomial vs fixed or bounded net weights for in-degrees, at least 0.78 for out-degrees). For young and multi-claw KCs, the fixed net weight had the highest evidence (Fig. 2b, c orange; log likelihood ratio at least 70.46 for fixed net weight vs bounded or binomial models on young KC in-degrees; at least 9.92 for multi-claw KC in-degrees; at least 48.17 for fixed net weight vs binomial on young KC out-degrees; at least 0.41 for fixed vs bounded net weight on young KC out-degrees; at least 201.8 for fixed net weight vs binomial on multi-claw KC out-degrees; at least 20.18 for fixed vs bounded net weight on multi-claw KC out-degrees). This suggests that less mature KCs have connectivity governed by a homeostatically regulated total input and output strength and as KCs mature, other factors come to dominate their wiring.

## 4 Discussion

We hypothesized that under a particular constraint, the probability of a neuron having degree $K$ is proportional to the size of the space of allowed circuit configurations with $K$ partners. This corresponds to the degree distribution of the maximum entropy synaptic weight configurations under a constraint. The general idea of considering the space of allowed configurations can be traced back to Elizabeth Gardner's pioneering work examining the storage capacity of the perceptron for random input patterns [15]. In the limit of infinitely many connections and input patterns, the idea that a neuron performs associations leads to predictions for the distributions of synaptic weights [3, 5, 2, 6]. Here, in contrast, we examined the hypothesis that the size of the space of allowed configurations governs the distribution of the number of connections. We examined constraints on the total strength of connections to (or from) a neuron and on individual connection strengths. The results with constraints on total connection strengths are a summary of results shown in more detail in [24].

**Connectivity constraints**  Previous studies have shown that minimizing the amount of wire used to connect neural circuits can predict the spatial layout of diverse neural systems (e.g., [12, 8, 21, 9, 7, 31, 4]) and pyramidal neurons' dendritic arborizations [10, 11]. Here we examined, in contrast, the idea that the number of synaptic partners to a neuron might be structured to make constraints flexible: to allow many different connectivity configurations under a constraint. We hope that this focus on models of synaptic weight configurations and degrees, rather than on physical wire and physical space, may expedite links with theories of computation.

We discussed constraints that limit or fix neurons' total input or output synaptic weight. We are not aware of experimental studies directly measuring synaptic scaling or resource limitations in Kenyon cells. There is evidence of homeostatic regulation of total synaptic weights in other *Drosophila melanogaster* neurons. Growth from the first instar larva to the third instar larva is accompanied by a homeostatic regulation of mechanosensory receptive fields [17] and functional motor neuron outputs [20] and nociceptive projections [16]. In addition, changes in inputs to the central aCC neuron elicit structural modifications to its dendritic tree that homeostatically maintain input levels [28].

**Regularization**  In machine learning, regularizing weights is a common way to reduce generalization errors. L2 regularization pressures the weights to lie in a L2-ball, and L1 pressures them to lie in an L1-ball; if the weights are also only positive, L1 regularization pressures weights to lie on the surface of a simplex. We examined regularization on its own, and observed that the sizes of solution spaces for simplicial weight constraints depends on the degree. This motivated us to consider cost functions (equivalently, probability distributions) for the degrees. We hope that these biologically inspired cost functions for connectivity degrees might be useful for architecture search.

**Computational capacity**  The Rademacher complexity of a set is bounded by its covering number: the number of spheres of radius $r$ that are required to cover a set [25]. The measure of configuration flexibility we used are Haussdorff measures of the solution spaces for different constraints. The Haussdorff measure has a similar flavor to the covering number. We have not yet formalized a relation between our approach and the Rademacher complexity, but believe this to be a promising direction.

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
