# OpenReview forum: "Flexible degrees of connectivity under synaptic weight constraints"
_NeurIPS.cc/2019/Workshop/Neuro_AI — Real Neurons & Hidden Units @ NeurIPS 2019 Poster_

### Official Review · AnonReviewer3 · 2019-09-22
**Interesting theory of homeostasis, constraints, and degree distributions; needs more discussion & AI intersection**

**Clarity:** 4

**Comment:**

How to structure a network in terms of in- and out-degrees of neurons is a fundamental question. In neuroscience, this area has been tackled more because neuron degree is easy to measure experimentally. On the AI side, there hasn't been as much focus on this kind of network structure, with the most common structures being fully-connected or convolutional. So I see this work as having potential relevance there. But more work will have to be done to see whether artificial neural networks end up following these same kind of principles.

I like the simplicity of the analysis in this work and the dataset that it is applied to. I only wish there were more discussion of the take-aways for both neuroscientists and AI researchers. But I see that interest for more as a positive rather than a negative.

**Category:**

Common question to both AI & Neuro

**Clarity Comment:**

The paper is well-written and for the most part easy to read. As mentioned in the technical review, the language of "size" should be made clearer in the introduction by stating that "size" will mean volume or area under different assumptions. Similarly, using the precise language of "degree" is preferable in my opinion to using "partners".

I would also spend a little more time in the intro motivating why "size" is something that'd be optimized. Can you learn more with a larger size, and do you think this connects to measures of dimensionality or complexity in ML systems (Rademacher/VC dim)? Also, constraining the degree is kind of what happens with convolutional layers, although they are very non-random.

There is no discussion section and there should be. What are the take-aways from this analysis? How do we interpret the conclusion that "other factors come to dominate" (L. 118) the network as Drosophila develops? I would like more speculation, for the biologists. Similarly, your model predicts an optimal K* dependent on various parameters; whereas this is known for mushroom body to be ~7 (in cerebellum, arguably similar, it is ~4). I'd like some discussion of whether your model is predictive of these properties or what it says about those networks' computational ability.

* L. 3 "size" -> "sizes" & "determines" -> "determine"
* L. 6 "partners" -> "neighbors" sounds better to me
* L. 17 "learning rule" is "not known", but what about Hebb/anti-Hebb STDP rules?
* L. 20 "consider the hypothesis" -> "hypothesize" would be better
* L. 25 "regulate" is used twice, I'd change the second to "stabilize" or "normalize" or similar
* L. 34 "We find that overall, ..." -> "We find that, overall, ..."
* L. 36 would be nice to have some speculation about this "developmental progression"
* L. 39 Suggest rephrase to "where a neuron has degree K" since you've already introduced degree = # neighbors = # partners
* L. 54 for balanced references, I'd add ref to recent work of Arenas on experimental verification of this scaling
* L. 60 "measureable synaptic weight changes" could mention "i.e., # of discrete vesicles"
* L. 69 "different types" I think you mean "many types" of neurons <- plural
* Figure 1: Suggest adding "bounded net", "fixed net", "bounded individual" labels to each row, on the left hand side under (a), (d), and (g)
* L. 90 "provides a cost function" is awkward, maybe simply "determines" is better
* L. 118 "Other factors come to dominate their wiring". What would these be? Since binomial is a good fit, would you say the network is random or not?
* L. 74 "net excitatory" strike "excitatory", you haven't talked about E/I at all so this is confusing

**Evaluation:**

4: Very good

**Importance:**

4: Very important

**Importance Comment:**

The authors have studied the problem of how the degree of neurons in a network influences their ability to learn. The idea is that degrees are less flexible than the weights of connections. Therefore, for a neuron with fixed degree, the "size" of the space of possible weights should be maximized. The size is computed 3 different ways, and this model is applied to the Drosophila mushroom body connectome. This is a fresh approach and has implications for AI; unfortunately they are not emphasized.

**Intersection:**

3: Medium

**Intersection Comment:**

I think this is the weakest part of the submission. The motivation for this work is almost entirely from the biological perspective. I think that this work probably does have some implications for AI, but it needs to be discussed by the authors. Places for this are in the introduction & potentially the discussion if added. (I am actually uncertain whether this workshop offers opportunity for revision, but I am writing my review like I would any paper and hope the authors will at least consider making some changes for their next version.)

To an AI person, familiar with statistical learning theory, it will probably be hard to find a good take-away from this work. I think a natural connection to try and make would be to the complexity of learning with such a network. Constraining the weights in a network to lie within some ball of radius R is a way to bound the generalization error. The idea of constraining the weights in a network is closely connected to classical types of regularization, which bound the norms of the weights. Another possible connection to illuminate would be to weight normalization techniques including batch normalization. Pruning neural networks to reduce their size is another area to look into, since that also reduces the degrees from fully-connected.

**Rigor Comment:**

I think the mathematics are correct and well-explained through Section 2. I thought the section on maximum entropy was harder to follow, probably because the inherent mathematics is more complicated. This could be alleviated by adding references & maybe pointing to an appendix. I would not care if the references extend beyond 4 pages; you can also eliminate line spacing there. You can also gain space by using the \paragraph commands instead of \subsection for the parts of Section 2.

Specific suggestions:
* L. 39 strike "synaptic partners" and use "degree"
* L. 40, I'd add "J_i \geq 0" when defining J_i.
* The volume & area of the simplex (ll. 58 & 73) need references.
* Notation S_K is not defined (l. 92), but I gather it is the volume/area calculations from before. I would suggest using S_K^{\leq net}, S_K^{= net}, and S_K^{individ} or something along those lines to clarify that these are the different ways of measuring "size". In fact, the language of "size" throughout the paper is kind of confusing until Sec. 2.1 when things become concrete. I would mention in the intro that you will use volume/area as ways to measure size.
* Ll. 67 & 79 "and vice versa" It is unclear to me what the vice versa case is. Clarify.
* L. 93 "K^max" isn't defined, strike "for large K^max"
* L. 99 what is \bar S? How you use the weights is muddled.
* L. 99 "Laplace approximation" and "model evidence" need references. I gather that "model evidence" is something like log-likelihood; be more precise.
* Ll. 101-102 in the binomial random wiring model, how do weights of the connections enter?


**Technical Rigor:**

4: Very convincing

---

> ### Comment · ~Gabriel_Koch_Ocker1 · 2019-10-31
> **Response**
>
> Thanks for the comments, support and the good suggestions. More details and discussion can be found in our preprint (https://doi.org/10.1101/603027).
>
> The Rademacher complexity of a set is bounded by its covering number. The measure of configuration flexibility we used are Haussdorff measures of the solution spaces for different constraints. The Haussdorff measure has a similar flavor to the covering number. We haven't yet formalized a relation between our approach and the Rademacher complexity, but agree that this would be a promising direction. One thing that our approach is related to is the simple bound on mutual information between inputs P and weights J obtained by neglecting the condition entropy H(J | P). The marginal entropy H(J) is given by the log size of a constraint's allowed connection space, so maximizing the size of the constraint space bounds the mutual information between inputs and connections.

---

### Official Review · AnonReviewer2 · 2019-09-25
**Difference in KC connectivity between young and adult flies**

**Clarity:** 3

**Category:**

Not applicable

**Clarity Comment:**

The paper is nicely written, but it reads like it was originally a much longer paper compacted down to four pages. For example in Section 3, there isn't much detail on the maximum entropy and Laplace approximation methods, or how the connectomic data was integrated into the analysis. Also, Fig. 2 could use some sort of adjustment to separate out the blue and orange lines.

**Evaluation:**

3: Good

**Importance:**

2: Marginally important

**Importance Comment:**

This paper addresses questions that are important to fly olfaction research, but the discussion doesn't say much about how it can provide insight for AI research.

**Intersection:**

2: Low

**Intersection Comment:**

Right now this finding falls mostly into neuroscience, and I feel this paper needs some added discussion on how the study of fly olfaction can be used to advance AI.

**Rigor Comment:**

The methodology present in the paper appears fine, but I have some questions about the neuroscience aspects:
- Is there any experimental evidence that KC synapses undergo synaptic scaling?
- Is there any experimental evidence that young flies have non-random KC connectivity?
- What was the threshold for determining significance between models? I ask because it seems the evidence is very close between the binomial model (which is experimentally suggested) and bounded/fixed net weights for single-claw adults, and a small change in what it means to be significant would lead to the result that synaptic connectivity in adults is also better supported by fixed/bounded weights.

**Technical Rigor:**

3: Convincing

---

> ### Comment · ~Gabriel_Koch_Ocker1 · 2019-10-31
> **Response**
>
> Thanks for the comments! We have revised the introduction and added discussion sections, pointing towards connections with architecture search, regularization and capacity bounds.
>
> We have not found any studies examining synaptic scaling in Kenyon cells, but do know of evidence for homeostatic regulation of connectivity in other Drosophila neurons. In D. melanogaster, growth from the first instar larva to the third instar larva is accompanied by a homeostatic regulation of mechanosensory receptive fields (Grueber et al 2009) and functional motor neuron outputs (Keshinian et al 1993) and nociceptive projections (Gerhard et al 2017). In addition, changes in inputs to the central aCC neuron elicit structural modifications to its dendritic tree that homeostatically maintain input levels (Tripodi et al 2008).
>
> Eichler et al. (2017) examined the structure of the 1st instar larva Kenyon cell wiring in some detail, but focused largely on clawed Kenyon cells. I don't think young KCs don't receive PN input, but there could be correlations in their recurrent connectivity. It would be interesting to examine the impact of young KCs on odor classification (if any); I'm not aware of a study looking at that.
>
> More detail on the approach and calculations can be found in the new supplemental material and our preprint (https://doi.org/10.1101/603027). The log odds for the various model comparisons shown in Figure 2 are stated in the text (last paragraph of the results section). There was no explicit significance cutoff in our Bayesian model comparison. We consider log odds around 1 to be weak evidence for one model over another and log odds of ~10 or more quite strong evidence.

---

### Official Review · AnonReviewer1 · 2019-09-26
**Statistical analysis of the olfactory wiring diagram with not much of biological insights**

**Clarity:** 2

**Comment:**

Because PNs-to-KCs connections are arguably not plastic, I'm not sure an analysis based on the weight volume is biologically relevant, even if the combinatorial term is added up to the model. Still I believe this line of work is important for understanding the wiring principle of the brain.

**Category:**

Common question to both AI & Neuro

**Clarity Comment:**

If you naively apply the stirling’s approximation, you get slightly different expression for Eqs (2) and (3). The author(s) should clarify the approximation they used.

I couldn’t get how the results shown in Fig. 2 are calculated either. In particular, the accuracy of the binomial model highly depends on whether only the connected pairs were fitted or all the potential connections were considered. Moreover, in the former scenario, the distribution needs to be shifted by one to cancel the selection bias. Yet, these points were not discussed in the manuscript.

**Evaluation:**

3: Good

**Importance:**

3: Important

**Importance Comment:**

In this work, the author(s) first characterized how various homeostatic constraints influences the optimal connection degree, then studied whether the connectivity structure found in Drosophila olfactory circuit is consistent with those homeostatic constraints. The results suggest that the model with the constraint on the net weight provides better fit for immature KCs than the  binomial wiring model. Although the work has some merit, I’m not convinced of the biological relevance.

**Intersection:**

3: Medium

**Intersection Comment:**

The choice of regularization is still an important topic in ML, and I believe it is insightful to study what kind of regularizer is used in the brain.

**Rigor Comment:**

In the manuscript, the parameter space was defined as a K-dimensional space, and the authors optimized K under some homeostatic constraints. However, considering the actual neural circuit, the problem should be defined as the problem of choosing (linear) K-dimensional subspace from N potential space (see eg. Ashok-kumar et al., Neuron, 2017). Thus, the benefit of having large K is underestimated in this study, which somewhat weakens the biological relevance.

In addition, the author(s) should discuss the range of p and J for which the optimal degree actually exists, as Eqs. (2) and (3) don’t have any (real) solution at wide range of p and J.

**Technical Rigor:**

2: Marginally convincing

---

> ### Comment · ~Gabriel_Koch_Ocker1 · 2019-10-31
> **Clarification**
>
> Thanks for the comment and suggestions! We'd like to clarify that we're considering a different problem than Litwin-Kumar et al. They sought to maximize the linear dimensionality (participation ratio) of responses under constraints on the number of inputs, number of neurons, or total number of connections---motivated specifically by circuits where an intermediate layer seems to randomly expand inputs before classification, such as the mushroom body. We asked what number of inputs maximizes the number of allowed synaptic weight configurations under different constraints (or rather for continuous weights, the measure of the allowed configuration space). We tested our results using data from the fly Kenyon cells, but don't think the idea is not specific to cerebellar-like systems or randomly expanding circuits. Rather, the basic motivation is the idea that changing the number of connections may be biologically more expensive than changing synaptic weights. We hope to link this more explicitly with bounds on computational capacity for small networks in future work.
>
> More details on the calculations and results can be found in our preprint (https://doi.org/10.1101/603027).

---

### Decision · Program_Chairs · 2019-10-02

Accept (Poster)